# The impact of epidemic infectious diseases on the relationship between subjective well-being and social class identity in older adults: The mediating role of Self-rated health

**Qianxi Feng**[1,2], **Yan Li**[1,2], **Miao Wan**[1,2], **Wei Li**[3]*

**1** School of Public Health, Chongqing Medical University, Chongqing, China, **2** Research Center for Medical and Social Development, School of Public Health, Chongqing Medical University, Chongqing, China, **3** The First Affiliated Hospital of Chongqing Medical University, Chongqing, China

* QX15982588754@outlook.com

## Abstract

### Background

The purpose of this paper is to explore the relationship between subjective well-being, social class identity, and Self-rated health among older persons,. Focusing on the mediating role of health and the impact of epidemic infectious diseases on these relationships.

### Methods

Based on the 2018 and 2021 China General Social Survey (CGSS) databases, the data were screened, and processed. Using Stata17, we employed ordered probit regression to examine the relationships among variables and Bootstrap methods to assess mediation effects, and the CGSS data for 2018 and 2021 were compared and analyzed.

### Results

Our results revealed that factors such as social class identity, health status, and personal income significantly positively impact older persons' subjective well-being (P<0.01). Notably, there was a partial mediating effect of health status between the subjective well-being of the elderly and social class identity. And findings showed that when older adults were affected by epidemic diseases, their subjective well-being, social class identity, and Self-rated health remained significantly positively correlated. Subjective well-being, social class identity. What is more noteworthy is that when affected by epidemic infectious diseases, older adults' subjective well-being, social class identity, and Self-rated health remained significantly positively correlated. The mediating role of self-rated health in older adults' subjective well-being and social class identity increased from 9.6% to 12.4%.

### Conclusions

In the face of epidemic infectious diseases, we need to pay more attention to the Self-rated health of the elderly, and the Chinese government should take effective measures to

**Data Availability Statement:** All relevant data are within the manuscript and its Supporting Information files.

**Funding:** The author(s) received no specific funding for this work.

**Competing interests:** The authors have declared that no competing interests exist.

improve their health level, which will in turn improve the subjective well-being of the elderly and realize the goal of healthy aging.

## Introduction

With the accelerating trend of global population aging, which has led to an increase in aging-related health complications and a growing demand for healthcare resources and diversification of healthcare needs, the health of the elderly has received increasing attention from society [1].Notably, epidemic infectious diseases pose a serious and persistent threat to the health of the elderly [2]. Healthy aging has become one of the hotspots in the field of public health. Subjective well-being (SWB) is a centralized expression of the overall spiritual life condition of the elderly [3], and it is one of the important indicators of Self-rated health and quality of life of the elderly [4].Relevant studies have shown SWB predicts psychological disorders such as depression in older adults [3]. At this stage, most of the researches explore the factors affecting the SWB of the elderly from both individual and social aspects. Individual factors mainly include marital status, residential status, Self-rated health, and annual income [5,6]; social factors mainly include social capital, social support, socioeconomic status, and living environment [7,8]. These factors affect SWB of older adults from both physical and psychological aspects. However, there are no studies that have examined SWB, social class identity and Self-rated health in depth in older people, and no studies that have explored whether Self-rated health plays a role in the relationship between SWB and social class identity in older people. Exploring the relationship between SWB, social class identity, and Self-rated health among the elderly and analyzing the role of health status in the SWB and social class identity of the elderly is of great significance for improving the quality of life of the elderly. At the same time, by analyzing the changes in the relationship between SWB, social class identity and Self-rated health during the outbreaks of epidemic infectious diseases and the changes in the mediating role of health status By exploring this, we can better understand the changes in the living status and psychological needs of older adults, thereby providing guidance for the development of effective health and psychological interventions, and making targeted recommendations for improving the SWB of older adults and achieving the goal of healthy aging.

Social class identity is the subject's perception of the position he or she occupies in the social class structure, which in turn constitutes the subjective class position [9]. Relevant studies show that older people with higher social class identity have higher SWB [10], and is an important factor affecting the well-being of the elderly [11]; scholars analyze the reasons for this, firstly, the subjects with higher social class identity are more satisfied with their lives, which can make their SWB at a higher level [12]; secondly, the groups with higher social class identity have relatively rich social resources compared with lower groups, and have more space for choice and less self-control. choice space and less self-control, thus forming a higher SWB with lower threat sensitivity [13]. Finally social comparison theory suggests that SWB is an evaluation formed by individuals by comparing their current situation [14], and that social class identity is also formed through comparison. Therefore, hypothesis H1 is proposed: social class identity can have a significant positive effect on the SWB of the elderly, i.e., the higher the social class identity the higher the SWB of the elderly has.

Self-rated health is an individual's subjective evaluation of his or her health, and its results are strongly consistent with objective health levels, making it one of the most reliable indicators of group health [15]. The sociology of health recognizes that social class identity is the root

cause of individual health differences. Compared with the elderly with high social class identity, those with low social class identity have fewer social resources, which makes it difficult for them to continuously meet their own health needs, and therefore it is more difficult for them to improve their Self-rated health [16]; moreover, the elderly with low social class identity do not have as good a living environment, social capital, and emotional support from their families as the elderly with high social class identity [17], which may easily lead to the decline of their health status [18,19]. Relevant studies show that individuals from the lower social classes have lower health indicators than those from the higher social classes [20]. Therefore, hypothesis H2 is proposed: social class identity can have a significant positive effect on the Self-rated health of the elderly, i.e., the higher the social class identity, the better the Self-rated health of the elderly.

Self-rated health is an important factor affecting the SWB of the elderly and plays a significant and stabilizing role in it [21]. Elderly people with high SWB have a higher chance of having good health; and mental and physical health have a close relationship with SWB [22], so improving the mental and physical health of the elderly can improve the SWB of the subject; some studies also show that good Self-rated health can effectively improve the SWB of the residents. According to related research, there is an interaction between SWB, social class identity and Self-rated health, i.e., older people with higher social class identity have higher life satisfaction and lower psychological burden, and a happy life state helps older people have healthier physiology and psychology, and older people can improve their SWB through better Self-rated health [23]. In addition, older people are more susceptible to psychological states such as depression, loneliness, neuroticism, and extraversion than people of other ages [24]. And some studies have confirmed that there is a correlation between subjective well-being, social class identity, and Self-rated health [4].Therefore, hypothesis H3 is proposed: Self-rated health plays a mediating role in the relationship between social class identity and SWB of the elderly.

Relevant studies have shown that epidemic infectious diseases can negatively affect both the physical and psychological aspects of older adults [25]. Psychological stress is caused by epidemic infectious diseases [26], and subjective well-being combined with hope and resiliency can effectively predict stress associated with pandemic infectious diseases [27,28]. In addition, this study compares and analyses the CGSS databases in 2018 and 2021 to investigate the effects of epidemic infectious diseases on the subjective sense of well-being, social class identity, and self-assessed health of older adults, as well as to analyze whether there is a mediating effect of self-assessed health on the phenomenon of growth in social class identity and SWB. Therefore, Hypothesis H4: The mediating effect of Self-rated health grows gradually in social class identity and SWB of the elderly.

This study explores the relationship between SWB, social class identity, and Self-rated health, analyzes whether health plays a mediating role in the relationship between social class identity and SWB among older adults, compares the mediating effect produced by the health status of older adults in the CGSS database in 2018 versus 2021, and explores whether the incidence of prevalent infectious diseases affects this mediating effect, It can better provide targeted suggestions for improving the SWB and social class identity of the elderly from the perspective of their Self-rated health. Based on the results of the data analysis and discussion, we will gain a deeper understanding of the SWB of the elderly and provide a scientific basis for improving the quality of life of the elderly, to promote society's concern for the SWB of the elderly, and to facilitate the rational allocation of social resources and the formulation and implementation of relevant policies to achieve the goal of healthy aging.

## Methods

### Data

In this study, in order to analyze the impact of epidemic infectious diseases on the subjective well-being, social class identity and Self-rated health of older adults, data from the 2018 and 2021 China General Social Survey (CGSS) were therefore selected. The survey is a large-scale national social sample survey initiated by the Renmin University of China and the Hong Kong University of Science and Technology, with the main purpose of understanding the living and employment conditions of urban and rural residents in China and their attitudes toward hot social issues. The survey is aimed at exploring the living conditions, employment situation, and attitudes of urban and rural residents towards hot issues in society, and it covers 30 provincial administrative regions in China, reflecting the real situation of our society in a more comprehensive and integrated way, and is scientific and representative. By analyzing these data, the study can reflect the subjective well-being, subjective class identity, and Self-rated health of the elderly more realistically and scientifically, which can provide strong support for our in-depth study of the relationship between subjective well-being, social class identity, and Self-rated health, and meets the needs of this study.

The CGSS was conducted in accordance with the ethical principles of the Declaration of Helsinki. The Ethics Committees of Renmin University of China and Hong Kong University of Science and Technology are responsible for ethical approval and consent to participate. We received authorization to use the publicly accessible CGSS. This study was conducted based on de-identified publicly available CGSS data, which is available at http://cgss.ruc.edu.cn/. Therefore, ethical approval or informed consent was no longer required for this study.

To exclude the interference of other factors, inclusion criteria of the data samples in this study were: age over 59 years old. The exclusion criteria were: (1) "Don't know" and "Refused to answer" values for the required variables (2) missing values for the required variables. By processing the data of the initial sample in the above way, 3717 (2018) and 1544 (2021) valid samples were finally included, and the data were correlated and analyzed in this study using stata17 software. Samples with missing values have been excluded from the analysis in this study.

### Variables selection

**Explained variable.**   The subjective well-being of the research sample is selected as the explanatory variable in this study. In China's research on residents' happiness, many scholars use the CGSS questionnaire "Overall, do you think you are happy in your life?" as the subjective happiness of the sample for research. Therefore, this question was chosen to measure the subjective well-being of the study sample by assigning the response options "very unhappy, relatively unhappy, can't say whether I am happy or not, relatively happy, and very happy" as 1, 2, 3, 4, and 5, in that order.

**Explanatory variable.**   The social class identity of the research sample was selected as the explanatory variable in this study. In academic research, the CGSS questionnaire "Generally speaking, to which social class do you belong in the current society?" This question is commonly used to measure the social class identity of the study sample and the CGSS database covers a large and wide sample size. Therefore this question provides a more accurate and scientific reflection of the social class identity of the study sample. The response options for this question were from 1 to 10 levels, where 1 is the lowest level, and 10 is the highest level, and the higher the level, the higher the social class identity of the elderly.

**Mediating variable.**   The mediating variable in this study was the Self-rated health of the study sample. Self-rated health is an individual's subjective evaluation of his or her health

status, and its results are consistent with the objective health status, which is one of the reliable indicators of an individual's health status. However, the corresponding question in the CGSS questionnaire, "What do you think is your current Self-rated health?" can accurately reflect the Self-rated health status of the study sample. Therefore, it was chosen to measure the Self-rated health of the study sample with the values of 1, 2, 3, 4, and 5 for the response options "very poor, poor, fair, good, and very good" respectively.

**Control variables.**   Previous studies have concluded that many factors affect an individual's SWB. Based on the published relevant studies, this study concludes that gender, household registration, annual personal income, education level, social trust, and social equity are the significant factors affecting subjective SWB. Therefore, these types of variables will be used as control variables in this study, in which male is assigned a value of 1 and female was assigned a value of 2. Agricultural hukou is assigned a value of 1, and non-agricultural hukou is assigned a value of 2. Uneducated is assigned a value of 0, and the education level of elementary school, junior high school, senior high school, university college, undergraduate and postgraduate students and above were assigned a value of 1, 2, 3, 4, 5, 6, and 7 in that order.Social trust and social fairness will be answered to the options in that order Assigned a value of 1, 2, 3, 4, 5, the larger the value indicates that the research participants recognize the social trust and social justice to a higher degree. This study will take the logarithm of the individual's annual income.

## Methods

This paper carries out correlation and mediation effect tests on the research samples, with P<0.05 as the difference is statistically significant. The sequential test method is a common method for conducting the mediation effect test. This study adopts the sequential test method to test the mediating effect on the research samples, and the flow of the sequential test method is shown in Fig 1. In addition to detecting the mediating effect of the research variables with the sequential test method, this study adopts the Bootstrap test method to test the mediating effect, which can be used as a supplement to the sequential test method to make up for the fact that the sequential test method is prone to endogeneity errors.

## Results

### Descriptive analysis

According to the descriptive statistics analysis, it can be seen that in the elderly group of the CGSS database in 2018, the proportion of males is 47.90%, the proportion of females is 52.10%, and the proportion of samples whose hukou is agriculture is 50.54%, as far as the education level is concerned, more than half of the study samples have not received education or their qualifications are elementary school, and 2.81% of the study samples have been educated with bachelor's degree and above. Among the older age groups in the 2021 CGSS database, the proportion of males and females is the same, the proportion of samples with non-agricultural households is 42.62%, the proportion of study samples with no education or with education in primary or junior high school is 70.00%, and 2.12% of study samples have education of bachelor's degree and above.

According to Table 1, it can be seen that among the main research variables, except for the self-assessed health status, it's than two variables' mean value increased, especially the social class identity, which increased by nearly 0.2. Specifically, compared with 2018, the elderly group in 2021 chose the health status as very unhealthy ratio increased by 2.81%, which made the mean value decreased; in terms of the elderly SWB, the 2021 percentage of older adults choosing very happy increased by 7.07% compared to 2018; in the variable of social class identity, the percentage of older adults choosing a rank higher than value 8 increased by 3.40% in

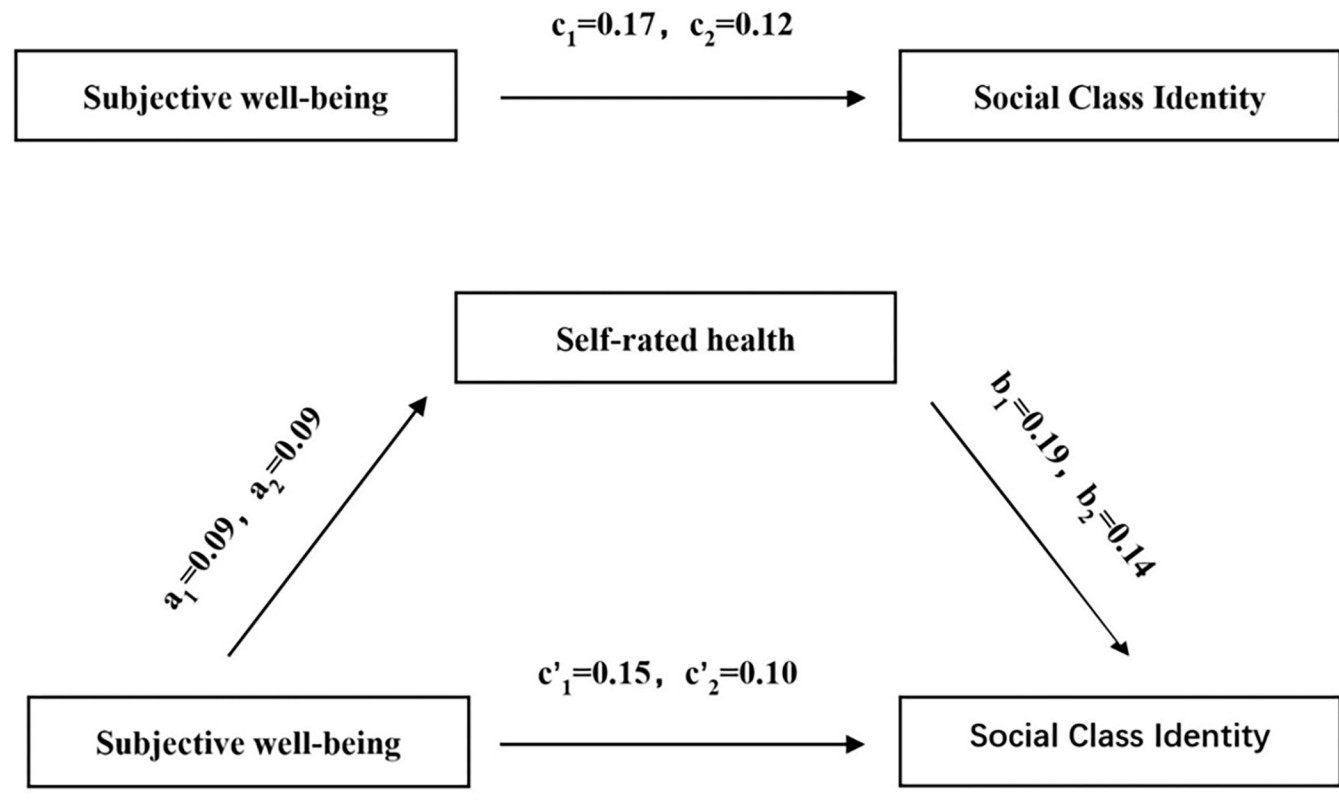

**Fig 1. Mediating effect model.**

2021 compared to 2018. The increase in the subjective happiness and income of the elderly may make their social class identity rank higher.

## Correlation test of main variables and results

Using Stata17 to test the correlation of the main variables in this study, it can be seen from Table 2 that SWB, social class identity, and health status show a positive correlation, which is statistically significant ($P<0.001$).

**Table 1. Results of descriptive statistical analysis of the study sample.**

| Variable | 2018 | | | | 2021 | | | |
|---|---|---|---|---|---|---|---|---|
| | **Mean** | **Std. dev.** | **Min** | **Max** | **Mean** | **Std. dev.** | **Min** | **Max** |
| subjective well-being | 3.961 | 0.828 | 1 | 5 | 4.059 | 0.850 | 1 | 5 |
| Social Class Identity | 4.159 | 1.721 | 1 | 10 | 4.291 | 2.010 | 1 | 10 |
| health | 3.118 | 1.052 | 1 | 5 | 3.110 | 1.112 | 1 | 5 |
| gender | 1.521 | 0.500 | 1 | 2 | 1.499 | 0.500 | 1 | 2 |
| census | 1.495 | 0.500 | 1 | 2 | 1.426 | 0.495 | 1 | 2 |
| education | 2.498 | 1.283 | 1 | 7 | 2.530 | 1.175 | 1 | 7 |
| ln-income | 9.456 | 1.452 | 4.605 | 15.42 | 10.14 | 2.357 | 3.912 | 16.12 |
| trust | 3.702 | 0.955 | 1 | 5 | 3.843 | 0.977 | 1 | 5 |
| fair | 3.326 | 1.017 | 1 | 5 | 3.612 | 0.999 | 1 | 5 |

**Table 2. Correlation test of key variables between 2018 and 2021.**

| Variable | 2018 | | | 2021 | | |
|---|---|---|---|---|---|---|
| | subjective well-being | Social Class Identity | Self-rated health | subjective well-being | Social Class Identity | Self-rated health |
| subjective well-being | 1 | | | 1 | | |
| Social Class Identity | 0.315*** | 1 | | 0.286*** | 1 | |
| Self-rated health | 0.235*** | 0.184*** | 1 | 0.215*** | 0.210*** | 1 |

Standard errors in parentheses.

* p < 0.05

** p < 0.01

*** p < 0.001.

## Mediating effects and results

To analyze whether there is a mediating effect between the correlation of each research variable and health status between the SWB and social class identity of the elderly, this study adopts the ordered Probit regression model with the bootstrap test to analyze the results, and this analysis includes four models. At the same time, this study compares the analysis results of the CGSS database in 2018 and 2021 to explore the changes in the correlation between the research variables and the mediating effect of health status in the two-year database. From the analytical results of Model I, it can be seen that the subjective class identity of the elderly in both 2018 (B = 0.172, P<0.001) and 2021 (B = 0.123, P<0.001) had a significant positive effect on SWB at the 1% level, i.e., therefore, Hypothesis H1 is valid. Model II showed a significant positive effect of social class identity on individual health status (B = 0.090, P<0.001), (B = 0.096, P<0.001) between the two-year study samples, i.e., hypothesis H2 was established. In Model III, the social class identity of the elderly group in the two-year database continues to have a significant positive impact on the SWB of the elderly (B = 0.159, P<0.001), (B = 0.110, P<0.001), and the regression coefficient declined by 0.02 over the two years; the mediating variable, health, is a significant positive impact on the SWB of the SWB of the elderly at a level of 1% to satisfy the conditions for the mediating effect of the sequential test method to hold. Therefore, it can be concluded based on the results of the three models that health status plays a partial mediating role in the relationship between social class identity and SWB of the elderly in the research samples of 2018 and 2021, and hypothesis H3 is established. Specifically, see Table 3. Fig 1 shows the model of the mediating effect of health status between social class identity and SWB of the elderly in 2018 and 2021, where subscripts 1 and 2 are the values of 2018 and 2019, respectively.

The bootstrap test was applied to test the robustness of the mediating effect of Self-rated health. From the results of Table 4, it can be seen that there is a mediating effect of Self-rated health in social class identity and SWB of the elderly in 2018 and 2021, with 95% CIs of (0.093, 0.125) and (0.055, 0.100), respectively, and the mediating effect increases from 9.6% in 2018 to 12.4% in 2021, and hypothesis H4 is established. The confidence intervals of both the direct and indirect effects do not contain zero, and the results of the mediation effect test indicate that health status plays a partial mediating role in the process of the influence of social class identity on the SWB of the elderly. Therefore, there is a mediating role of health status between SWB and the social class identity of the elderly, and its mediating effect is increasing.

## Discussion

The results of the study show a significant positive correlation between SWB, social class identity, and health status. This positive correlation indicates that the higher the social class identity

**Table 3. Results of ordered probit regression models for the main variables of the study sample.**

| Variable | 2018 | | | 2021 | | |
|---|---|---|---|---|---|---|
| | subjective well-being | Self-rated health | subjective well-being | subjective well-being | Self-rated health | subjective well-being |
| Social Class Identity | 0.172*** | 0.090*** | 0.159*** | 0.123*** | 0.096*** | 0.110*** |
| | (0.012) | (0.011) | (0.012) | (0.015) | (0.014) | (0.016) |
| Self-rated health | | | 0.193*** | | | 0.149*** |
| | | | (0.019) | | | (0.028) |
| gender | 0.101** | -0.104** | 0.121** | -0.071 | -0.132* | -0.052 |
| | (0.038) | (0.036) | (0.039) | (0.060) | (0.055) | (0.060) |
| census | 0.027 | -0.054 | 0.037 | 0.175* | 0.103 | 0.159* |
| | (0.056) | (0.052) | (0.056) | (0.068) | (0.063) | (0.068) |
| education | 0.012 | 0.065*** | -0.000 | 0.010 | 0.104*** | -0.005 |
| | (0.018) | (0.017) | (0.018) | (0.029) | (0.026) | (0.029) |
| income | 0.090*** | 0.100*** | 0.072*** | 0.006 | 0.018 | 0.003 |
| | (0.020) | (0.018) | (0.020) | (0.013) | (0.012) | (0.013) |
| trust | 0.149*** | 0.025 | 0.147*** | 0.191*** | -0.015 | 0.195*** |
| | (0.020) | (0.019) | (0.020) | (0.031) | (0.029) | (0.031) |
| fair | 0.329*** | 0.056** | 0.325*** | 0.327*** | 0.086** | 0.318*** |
| | (0.020) | (0.018) | (0.020) | (0.031) | (0.029) | (0.031) |
| $N$ | 3717 | 3717 | 3717 | 1544 | 1544 | 1544 |
| $R^2$ | 0.1007 | 0.0251 | 0.1136 | 0.0938 | 0.0276 | 0.1024 |

Standard errors in parentheses.

* $p < 0.05$

** $p < 0.01$

*** $p < 0.001$.

of older adults, the stronger their SWB and better health status, which is the same as the results of previous studies. In addition, compared to 2018, older adults' economic income increased in 2021, and both SWB and social class identity increased. In addition, in 2018, income, trust, and public average positively predicted older adults' subjective well-being; however, in 2021, income was not strongly correlated with older adults' subjective well-being. Socioemotional Selectivity Theory is a theory of lifelong development of motivation proposed by Stanford University psychologist Laura Carstensen [29]. Socioemotional Selectivity Theory suggests that as the subject grows older, he or she recognizes the finite nature of the remaining life and gradually shifts the main goal of life: from the pursuit of intellectual motivation to the acquisition of emotional motivation [29]. Moreover, the arrival of pandemic infectious diseases makes the elderly pay more attention to their own health and less attention to income. As a result, the impact of income on subjective well-being is diminishing.

**Table 4. Self-rated health-mediated effects test.**

| Effect type | Path | 2019 | | | | 2021 | | | |
|---|---|---|---|---|---|---|---|---|---|
| | | Effect value | Bootstrap SE | Boot 95% CI | Amount of effect (%) | Effect value | Bootstrap SE | Boot 95% CI | Amount of effect (%) |
| Mediation effect | Social Class Identity → Self-rated health → SWB | 0.010 | 0.001 | 0.006, 0.013 | 9.6% | 0.010 | 0.002 | 0.004,0.014 | 12.4% |
| Direct effect | Social Class Identity → SWB | 0.099 | 0.007 | 0.083, 0.114 | 90.4% | 0.068 | 0.011 | 0.046,0.090 | 87.6% |
| Total effect | | 0.109 | 0.008 | 0.093, 0.125 | 10.9% | 0.078 | 0.011 | 0.055,0.100 | 7.8% |

In addition, the study found that social class identity has a significant positive effect on the SWB of older adults. Individuals' social class identity is formed both through group comparison and self-comparison [30]; especially in the context of China's socio-economic transformation, the intergenerational upward mobility of residents is high, and upward mobility will compare themselves with groups of a higher social class, which will allow people to identify themselves with a higher subjective social class, and is a way for them to have a greater sense of acquisition and fulfillment, which in turn powerfully enhances their SWB [31]. Based on this, to enhance the SWB of the elderly, we should promote the construction of activity venues for the elderly based on the location of large groups of people such as communities and village committees, and actively carry out cultural and sports activities and volunteer activities that meet the needs of the elderly, to expand the spare-time activities of the elderly, reduce their sense of isolation [24,32], and increase their life satisfaction and SWB, to promote the realization of the goal of healthy aging.

This study also found that social class identity has a significant positive effect on individual health status, i.e., individuals with higher social class identity ratings have higher self-rated health status. Related studies have shown that social class identity is associated with self-rated health and that individuals with higher social class identity have more economic and social resources to maintain or improve their health. Compared with the group with lower social class identity, the group with higher social class identity has more resources to satisfy their own health needs, and also can easily satisfy their own psychological needs, so they have higher social satisfaction and sense of acquisition and thus have better self-assessed health status [33]. Based on this, we should pay attention to the mental and physical health of the elderly, actively communicate with the elderly, and give them more care in both physical and psychological dimensions, to improve their physical and mental health and their sense of belonging to the family and sense of security, and thus enhance their social class identity.

This study shows that the mediating variable health status plays a positive and significant effect on the SWB of older adults and mediates between social class identity and the SWB of older adults. Moreover, the mediating effect of health status is increasing, i.e., social class identity contributes to the acquisition of SWB of older adults through health status. The mechanism is that older people with higher social class identity, firstly, have better social and economic conditions to improve their health status, which enhances their quality of life and increases their SWB; and secondly, they have more social resources to improve their quality of life, which improves their life satisfaction and SWB [34]. Maslow's Hierarchy of Needs is a motivational theme in psychology that includes five levels of human needs. The needs are as follows: physiological needs, safety needs, social needs, respect, and self-actualization [35]. Maslow's Hierarchy of Needs theory suggests that only after a certain level of needs is relatively satisfied will the subject pursue a higher level of needs. The improvement of the health status of the elderly means the improvement of the life and quality of life of the elderly, and they have more energy to satisfy their higher-level needs, thus improving their subjective well-being [35]. This study found that health status plays a partly mediating effect in the promotion of social class identity on SWB, i.e., health status has a "diversionary" effect between the two, firstly, the health status in SWB is essentially the degree of satisfaction of individuals with their Self-rated health status. The significant difference between social equity and social trust is a key factor affecting Self-rated health, so China should emphasize the construction of basic medical care in rural areas, shorten the gap between urban and rural medical facilities and medical service levels, and do a good job in urban and rural health publicity to improve the social and medical environment of the elderly, to improve the satisfaction of the elderly's Self-rated health, and then improve the subjective sense of well-being. In addition to SWB, the influence of health status on social class identity is a more complex issue. Based on the

improvement of the SWB of the elderly, the Chinese government should provide fairer and more accessible medical services and improve the social class identity of the elderly by promoting fair income distribution and fair education, to promote a higher sense of SWB among the elderly, and to promote the realization of Self-rated health aging goals. and promote the realization of the goal of healthy aging.

Finally, as far as the results of the comparative analysis based on the CGSS data of 2018 and 2021 are concerned, the SWB, social class identity, and Self-rated health of the elderly continue to show a significant correlation when subjected to epidemic infectious diseases. The impact of income on health status has been reduced by epidemic infectious diseases. In addition, the degree of interaction of the remaining variables decreased, except for social class identity, which had a slightly stronger influence on health status. Epidemic infectious diseases may directly reduce the health status of older adults [36,37], who are more susceptible to viral infections and serious medical conditions since they usually have lower immunity. In addition, if older people do not receive timely and appropriate medical care, their health conditions may deteriorate further, and epidemic infectious diseases may increase their psychological stress [38]. Elderly people with higher social class identity have more social and financial resources to relieve their psychological stress compared to those with lower social class identity [39], so the extent of social class identity's influence on health status is enhanced. Under the influence of epidemic infectious diseases, the mean value of self-assessed health status of the elderly became lower, and at the same time, the influence of SWB of the elderly from social equity became more, so the influence of health status on this was reduced. Moreover, during the period of epidemic infectious diseases, the influence of social trust and social fairness on the SWB of the elderly is strengthened,and social trust and social equity are important influences on the SWB of older adults. Social trust and social equity can influence the level of citizens' political participation, and a more active level of political participation can lead to higher social trust and social equity. In 2018 & 2021, due to the outbreak of epidemic infectious diseases, Chinese citizens have more opportunities for political participation, and therefore their social trust and social equity are enhanced, which more strongly influences the SWB of the elderly. while the influence of income on SWB decreases, Due to the influence of epidemic infectious diseases, Chinese citizens pay more attention to the sense of social fairness and social trust, which reduces the attention to income, income is an important factor affecting social class identity, so the influence of social class identity on SWB of the elderly becomes smaller. In addition to this, the mediating effect of Self-rated health under the influence of epidemic infectious diseases has increased. Based on this, it is crucial to improve the health level of the elderly, and the government and society should provide more medical insurance and health service resources, improve medical conditions and medical resource allocation, and improve the health level of the elderly. In addition, the elderly themselves should pay attention to personal hygiene and protective measures to reduce the risk of contracting viruses. In the face of an epidemic, the Government and society should take effective measures to reduce the social isolation and loneliness of the elderly. For example, they can organize plentiful social activities to provide psychological support and comfort, as well as strengthen the role of community and village committees to provide more care and support to the elderly. In addition, in the face of the epidemic, it is still important to enhance the social class identity of the elderly, so the Chinese government should narrow the gap between urban and rural medical conditions, promote income distribution and education equity, and improve social fairness and social trust. At the same time, older persons themselves should pay attention to their social class identity and actively participate in social activities and volunteer work to improve their social status and prestige.

## Conclusion

To summarize, there is a significant positive correlation between SWB, social class identity, and Self-rated health among the elderly, Self-rated health plays an intermediary role in SWB and social class identity, and its intermediary role has been strengthened under the influence of epidemic infectious diseases. Based on this, we need to pay more attention to the health status of the elderly and its impact on SWB and social class identity in the face of epidemic infectious diseases. The Chinese government should improve the health and social class identity of the elderly by taking effective measures, such as narrowing the gap between urban and rural medical conditions and enriching social activities, to improve the SWB of the elderly and achieve the goal of healthy aging.

There are still some shortcomings in this study, for example, this study analyzes the role of health status in social class identity and SWB of the elderly from a macro point of view, and this analysis needs a more accurate micro-interpretation model for more in-depth discussion. In addition, due to the differences in research time and personnel, the surveyed older adults in the CGSS database in 2018 and 2021 are not the same, which may lead to some bias and make the comparison results uncertain.

## Supporting information

**S1 File.**
(DOCX)

**S2 File.**
(ZIP)

## Acknowledgments

We thank Chinese General Social Survey for their excellent work in database design and data collection and for allowing free access to the data.

## Author Contributions

**Conceptualization:** Qianxi Feng, Yan Li, Miao Wan.

**Data curation:** Qianxi Feng.

**Formal analysis:** Qianxi Feng.

**Supervision:** Miao Wan, Wei Li.

**Writing – original draft:** Qianxi Feng.

**Writing – review & editing:** Qianxi Feng, Yan Li.

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
