## [Decision Letter · Decision Letter 0]

26 Nov 2023

PONE-D-23-34398The impact of epidemic infectious diseases on the relationship between subjective well-being and social class identity in older adults: the mediating role of Self-rated healthPLOS ONE

Dear Dr.Author,

Thank you for submitting your manuscript to PLOS ONE. After careful consideration, we feel that it has merit but does not fully meet PLOS ONE’s publication criteria as it currently stands. Therefore, we invite you to submit a revised version of the manuscript that addresses the points raised during the review process.

We look forward to receiving your revised manuscript.

Kind regards,

Roghieh Nooripour, Ph.D

Academic Editor

PLOS ONE

Journal Requirements:

“No conflict of interest”

4. PLOS requires an ORCID iD for the corresponding author in Editorial Manager on papers submitted after December 6th, 2016. Please ensure that you have an ORCID iD and that it is validated in Editorial Manager. To do this, go to ‘Update my Information’ (in the upper left-hand corner of the main menu), and click on the Fetch/Validate link next to the ORCID field. This will take you to the ORCID site and allow you to create a new iD or authenticate a pre-existing iD in Editorial Manager. Please see the following video for instructions on linking an ORCID iD to your Editorial Manager account: https://www.youtube.com/watch?v=_xcclfuvtxQ.

Additional Editor Comments (if provided):

Dear Author,

I hope this message finds you well. I have carefully reviewed your manuscript titled "The Impact of Epidemic Infectious Diseases on the Relationship between Subjective Well-being and Social Class Identity in Older Adults: The Mediating Role of Self-rated Health." Thank you for the effort and time you have put into your research.

After a thorough evaluation, it is my responsibility to provide feedback on the current status of your manuscript. I believe that your work has substantial potential, but it requires significant revisions to meet the academic standards and expectations of our journal.

I understand that revising a manuscript can be a challenging task, but I believe that addressing these issues will significantly enhance the quality and impact of your research. I encourage you to carefully consider these recommendations and make the necessary revisions.

Please feel free to reach out if you have any questions or need further clarification on any of the suggested revisions. Once you have completed the revisions, please resubmit your manuscript through our submission portal. I look forward to reviewing the revised version of your work.

Best regards,

Reviewers' comments:

Reviewer's Responses to Questions

**Comments to the Author**

1. Is the manuscript technically sound, and do the data support the conclusions?

Reviewer #1: Partly

Reviewer #2: Yes

2. Has the statistical analysis been performed appropriately and rigorously? 

Reviewer #1: Yes

Reviewer #2: I Don't Know

3. Have the authors made all data underlying the findings in their manuscript fully available?

Reviewer #1: Yes

Reviewer #2: Yes

4. Is the manuscript presented in an intelligible fashion and written in standard English?

Reviewer #1: Yes

Reviewer #2: Yes

5. Review Comments to the Author

Reviewer #1: I have thoroughly reviewed your paper and appreciate your efforts. I've given constructive feedback to improve your article's quality. Please consider my suggestions to enhance clarity and impact. Once you make revisions, I will gladly review the updated version. Your commitment to improvement is commendable, and I look forward to seeing how your article evolves.

Best regards.

Abstract

• Start the abstract with a clear and concise title that encapsulates the main focus of the study. The purpose of the study is well-articulated, but consider rephrasing it for more clarity. For instance, "This study explores the relationship between subjective well-being, social class identity, and self-rated health among older persons in China, focusing on the mediating role of health and the impact of epidemic infectious diseases on these relationships."

• The methods section is concise and mentions the use of the CGSS databases, Stata17 software, and specific statistical techniques. Consider adding a brief explanation of why these specific methods were chosen to enhance the reader's understanding. For example, "Using Stata17, we employed ordered probit regression to examine the relationships among variables and Bootstrap methods to assess mediation effects."

• This section effectively summarizes key findings. However, it could benefit from a clearer presentation. For instance, "Our results revealed that factors such as social class identity, health status, and personal income significantly positively impact older persons' subjective well-being (P<0.01). Notably, health status partially mediated the relationship between subjective well-being and social class identity. This med

Introduction

1. Provide a brief overview of global aging trends and their implications for public health. This sets the stage for why your study is timely and significant.

2. The definition of subjective well-being (SWB) is well-placed. Consider expanding on how SWB specifically pertains to the elderly population, perhaps referencing key studies that have demonstrated its importance in this demographic.

3. The introduction of individual and social factors influencing SWB is good. Elaborate slightly more on why these factors are crucial in understanding SWB among the elderly.

4. Clearly highlight the gap in current research, particularly the unexplored role of Self-rated health in the SWB-social class identity relationship among the elderly. This strengthens the rationale for your study.

5. Each hypothesis is well-founded and logically flows from the preceding discussion. Ensure that each hypothesis is distinctly stated and directly linked to the literature or theory it draws from.

6. For enhanced readability, consider formatting each hypothesis as a separate point or paragraph.

7. The discussion on the influence of Self-rated health on SWB is insightful. Expand on how this relationship might be unique or particularly pronounced in the elderly population, as compared to other age groups.

Method

• Clearly justify the selection of the CGSS 2018 and 2021 datasets. Explain why these specific years were chosen and their relevance to your study objectives.

• Elaborate on the criteria used to exclude invalid responses, extreme values, and missing data. This will help in understanding how you ensured the quality and reliability of your sample.

• Clearly define how each variable was operationalized. For the subjective well-being variable, explain why you chose that specific question from the CGSS and how it effectively represents SWB.

• Similarly, for social class identity and Self-rated health, justify the selection of these specific measures. Discuss any potential limitations or biases in self-report measures.

• Provide a rationale for the selection of each control variable. Explain how gender, household registration, personal income, education level, social trust, and social fairness are relevant to your study.

• Clearly describe how each control variable is measured and quantified.

• Justify the choice of correlation and mediation effect tests. Explain why these methods are appropriate for your research questions and hypotheses.

• Provide more details on the sequential test method and the Bootstrap test method. Explain how these methods complement each other and address any potential shortcomings.

Results

1. The descriptive statistics provide a clear picture of the sample characteristics. Consider briefly interpreting these findings, highlighting any notable changes or trends between 2018 and 2021. For example, the increase in the proportion of elderly individuals with higher social class identity could be discussed in terms of its potential implications.

Discussion

• Clearly articulate how your findings align or differ from previous research. Discuss the reasons behind the observed significant positive correlation between SWB, social class identity, and health status.

• Explain how the findings relate to the specific socio-economic context of China, considering the high rate of intergenerational upward mobility.

• Elaborate on why individuals with higher social class identity might have better self-rated health status. Discuss the potential psychological and material factors contributing to this finding.

• Suggest specific policies or interventions that could enhance the health and SWB of the elderly, based on your findings.

• Delve into the mechanisms through which health status mediates the relationship between social class identity and SWB. Discuss the implications of this 'diversionary' effect of health status.

• Suggest how public health initiatives could leverage these insights to improve elderly well-being.

• Discuss how the pandemic has specifically affected the elderly's health, SWB, and social class identity. Explain why these changes have occurred and what they mean for future public health strategies.

Reviewer #2: Hello and thank you for giving me the opportunity to read and review this valuable article. After reviewing the submitted version, some tips to improve the quality and better understanding of the readers are mentioned in the following section:

The subject of the article is unique and valuable. The results section is well explained.

1. The title of the article mentions infectious diseases, but it is not mentioned except in the conclusion, even in the introduction.

2. It seems that the title needs to be changed because it has been discussed in relation to the impacts of infectious disease, but in principle, the key role of the variable of self-rated health in the relationship between subjective wellbeing and social class identity has been discussed.

3. It is suggested to provide a definition of the topic or mention its importance in the background section (abstract) rather than simply repeating the purpose of the article.

3. The first three lines of the introduction have no reference.

4. In the introduction, you have mentioned that self-rated and its effect on the relationship between subjective wellbeing and social class identity have not been investigated. A question arises in the reader's mind that why this factor should affect the relationship between these two variables and is there a need for a research? It is suggested to explain its importance in one line.

5. Defining the main variables in the opening paragraphs makes the text better understood, so it is recommended to define the variables first and mention their importance in the introduction section, and then mention the importance of research about them.

6. It is not mentioned in the method section what age group is meant by elderly people and what age range is considered?

7. In the method section, it is not mentioned how the information was collected? on the phone? In person? Internet?

8. In relation to CGSS questionnaire, a brief explanation should be provided.

9. The elderly usually suffer from memory-related diseases. What solution did you think for this group of people? Were they identified and were the samples excuded?

10. It is suggested to explain the limitations in more detail.

6. PLOS authors have the option to publish the peer review history of their article (what does this mean?). If published, this will include your full peer review and any attached files.

Reviewer #1: No

Reviewer #2: No

---

## [Author Response · Author response to Decision Letter 0]

25 Jan 2024

Dear Editor,

Thank you for giving us an opportunity to revise our manuscript, we appreciate editor and reviewers very much for their positive and constructive comments and suggestions on our manuscript entitled “The impact of epidemic infectious diseases on the relationship between subjective well-being and social class identity in older adults: the mediating role of Self-rated health”(PONE-D-23-34398). We have studied the comments carefully and have made revision in this paper. The responses to the comments from three reviewers are offered separately, and the corresponding modification position page in the revised manuscript can be found in the rightmost column of the response table blow. Moreover, the revised paragraphs (sentences) are labeled with revision mode.

The paragraph or sentence revised according to the REVIEWER is represented by a revision pattern.

Respond to Reviewer #1’comments

1.Abstract

• Start the abstract with a clear and concise title that encapsulates the main focus of the study. The purpose of the study is well-articulated, but consider rephrasing it for more clarity. For instance, "This study explores the relationship between subjective well-being, social class identity, and self-rated health among older persons in China, focusing on the mediating role of health and the impact of epidemic infectious diseases on these relationships."

• The methods section is concise and mentions the use of the CGSS databases, Stata17 software, and specific statistical techniques. Consider adding a brief explanation of why these specific methods were chosen to enhance the reader's understanding. For example, "Using Stata17, we employed ordered probit regression to examine the relationships among variables and Bootstrap methods to assess mediation effects."

• This section effectively summarizes key findings. However, it could benefit from a clearer presentation. For instance, "Our results revealed that factors such as social class identity, health status, and personal income significantly positively impact older persons' subjective well-being (P<0.01). Notably, health status partially mediated the relationship between subjective well-being and social class identity. This med

Thank you very much for your expert opinion on our article abstract. As you may have noticed, several issues need to be addressed. Based on your valuable suggestions, we have made extensive changes to our previous abstract, described in the article's abstract section.

2.Introduction

• Provide a brief overview of global aging trends and their implications for public health. This sets the stage for why your study is timely and significant.

• The definition of subjective well-being (SWB) is well-placed. Consider expanding on how SWB specifically pertains to the elderly population, perhaps referencing key studies that have demonstrated its importance in this demographic.

• The introduction of individual and social factors influencing SWB is good. Elaborate slightly more on why these factors are crucial in understanding SWB among the elderly.

• Clearly highlight the gap in current research, particularly the unexplored role of Self-rated health in the SWB-social class identity relationship among the elderly. This strengthens the rationale for your study.

•Each hypothesis is well-founded and logically flows from the preceding discussion. Ensure that each hypothesis is distinctly stated and directly linked to the literature or theory it draws from.

•For enhanced readability, consider formatting each hypothesis as a separate point or paragraph.

• The discussion on the influence of Self-rated health on SWB is insightful. Expand on how this relationship might be unique or particularly pronounced in the elderly population, as compared to other age groups.

Thank you very much for your professional and detailed comments on the introduction. Based on your valuable suggestions, we have revised the introduction, such as adding the global aging trend and its impact on public health; expanding the specific relationship between subjective well-being and the elderly population, and strengthening the basis of the research hypothesis, etc., please refer to the introduction section in the artic

3.Method

• Clearly justify the selection of the CGSS 2018 and 2021 datasets. Explain why these specific years were chosen and their relevance to your study objectives.

• Elaborate on the criteria used to exclude invalid responses, extreme values, and missing data. This will help in understanding how you ensured the quality and reliability of your sample.

• Clearly define how each variable was operationalized. For the subjective well-being variable, explain why you chose that specific question from the CGSS and how it effectively represents SWB.

• Similarly, for social class identity and Self-rated health, justify the selection of these specific measures. Discuss any potential limitations or biases in self-report measures.

• Provide a rationale for the selection of each control variable. Explain how gender, household registration, personal income, education level, social trust, and social fairness are relevant to your study.

• Clearly describe how each control variable is measured and quantified.

• Justify the choice of correlation and mediation effect tests. Explain why these methods are appropriate for your research questions and hypotheses.

• Provide more details on the sequential test method and the Bootstrap test method. Explain how these methods complement each other and address any potential shortcomings.

Thank you sincerely for your comments on the Methods section. Based on your valuable suggestions, we have modified the methodology section, such as adding the rationale and exclusion criteria for selecting the 2018 and 2021 CGSS datasets and clarifying the definition and operational criteria for each variable. Besides, the rationale for choosing the correlation test and mediation effect test is placed in the introduction section.

4.Results

• The descriptive statistics provide a clear picture of the sample characteristics. Consider briefly interpreting these findings, highlighting any notable changes or trends between 2018 and 2021. For example, the increase in the proportion of elderly individuals with higher social class identity could be discussed in terms of its potential implications.

Thank you for your suggestions on the results, we have revised the results section as you suggested.

5.Discussion

• Clearly articulate how your findings align or differ from previous research. Discuss the reasons behind the observed significant positive correlation between SWB, social class identity, and health status.

• Explain how the findings relate to the specific socio-economic context of China, considering the high rate of intergenerational upward mobility.

• Elaborate on why individuals with higher social class identity might have better self-rated health status. Discuss the potential psychological and material factors contributing to this finding.

• Suggest specific policies or interventions that could enhance the health and SWB of the elderly, based on your findings.

• Delve into the mechanisms through which health status mediates the relationship between social class identity and SWB. Discuss the implications of this 'diversionary' effect of health status.

• Suggest how public health initiatives could leverage these insights to improve elderly well-being.

• Discuss how the pandemic has specifically affected the elderly's health, SWB, and social class identity. Explain why these changes have occurred and what they mean for future public health strategies.

Thank you very much for your suggestions on the discussion, we have revised the discussion section as you suggested. Your valuable suggestions are very important for us to revise and improve the discussion. It makes the article more obviously innovative and deepens the relationship between the variables in the specific socio-economic context of China. In addition, we have further discussed the mechanism by which health status mediates the relationship between social class identity and sector-wide obligations. Finally, we have placed comments at the end of each paragraph of the discussion. Thank you.

Respond to Reviewer #1’comments

1. The title of the article mentions infectious diseases, but it is not mentioned except in the conclusion, even in the introduction.

Thank you for your comments, we have taken the description of the increased prevalence of infectious diseases and written it in the introduction as well.

2. It seems that the title needs to be changed because it has been discussed in relation to the impacts of infectious disease, but in principle, the key role of the variable of self-rated health in the relationship between subjective wellbeing and social class identity has been discussed.

Thank you for your comment, we have noted this issue. For this reason, the discussion of epidemic infectious diseases has been increased throughout the text.

3. It is suggested to provide a definition of the topic or mention its importance in the background section (abstract) rather than simply repeating the purpose of the article.

Thank you for your comments, we have revised the research section in accordance with your comments.

4. The first three lines of the introduction have no reference.

Thanks to your comments, we have added references to the first three lines of the introduction.

5. In the introduction, you have mentioned that self-rated and its effect on the relationship between subjective wellbeing and social class identity have not been investigated. A question arises in the reader's mind that why this factor should affect the relationship between these two variables and is there a need for a research? It is suggested to explain its importance in one line.

Thanks to your suggestion, we have mentioned the relationship between subjective well-being, subjective class identity, and health status in older adults in the introduction.

6. Defining the main variables in the opening paragraphs makes the text better understood, so it is recommended to define the variables first and mention their importance in the introduction section, and then mention the importance of research about them.

Thank you from the bottom of our hearts for your suggestions, we have revised and improved the introduction according to your suggestions.

7. It is not mentioned in the method section what age group is meant by elderly people and what age range is considered?

Thank you from the bottom of our hearts for your suggestions, and we have refined the methodology according to your suggestions. We are following the World Health Organization's definition of older adults, and the sample aged 60 years and above was used as the population for this study - older adults.

8. In the method section, it is not mentioned how the information was collected? on the phone? In person? Internet?

Thank you for your advice. The CGSS public database we used for the study. The survey is a large-scale national social sample survey initiated by the Renmin University of China and the Hong Kong University of Science and Technology, with the main purpose of understanding the living and employment conditions of urban and rural residents in China and their attitudes toward hot social issues. The researchers of this data, all of whom have been professionally trained and instructed, administered the questionnaire to the respondents face-to-face.

9. In relation to CGSS questionnaire, a brief explanation should be provided.

Thank you for your comments, which I have added to the article.

The survey is a large-scale national social sample survey initiated by the Renmin University of China and the Hong Kong University of Science and Technology, with the main purpose of understanding the living and employment conditions of urban and rural residents in China and their attitudes toward hot social issues. The survey is aimed at exploring the living conditions, employment situation, and attitudes of urban and rural residents towards hot issues in society, and it covers 30 provincial administrative regions in China, reflecting the real situation of our society in a more comprehensive and integrated way, and is scientific and representative. By analyzing these data, the study can reflect the subjective well-being, subjective class identity, and Self-rated health of the elderly more realistically and scientifically, which can provide strong support for our in-depth study of the relationship between subjective well-being, social class identity, and Self-rated health, and meets the needs of this study.

10. The elderly usually suffer from memory-related diseases. What solution did you think for this group of people? Were they identified and were the samples excuded?

Thank you for your very valuable suggestions, according to this problem we have had a lot of discussions and reviewed the relevant literature, but we have not yet found a satisfactory method. This is also a deficiency of this study, we will conduct more in-depth research on this issue in the future, and hope to get your understanding.

11. It is suggested to explain the limitations in more detail.

Thank you very much for your comments, we have explained the limitations further.

---

## [Editor Report · Decision Letter 1]

1 Feb 2024

PONE-D-23-34398R1The impact of epidemic infectious diseases on the relationship between subjective well-being and social class identity in older adults: the mediating role of Self-rated healthPLOS ONE

Dear Dr. Wei,

Thank you for submitting your manuscript to PLOS ONE. After careful consideration, we feel that it has merit but does not fully meet PLOS ONE’s publication criteria as it currently stands. Therefore, we invite you to submit a revised version of the manuscript that addresses the points raised during the review process.

We look forward to receiving your revised manuscript.

Kind regards,

Academic Editor

PLOS ONE

Additional Editor Comments:

Introduction

Elaborate more on the broader significance and implications of the study in the context of healthy aging. Highlight the potential contributions of the research to the existing literature and how it addresses gaps in understanding the relationship between SWB, social class identity, and Self-rated health in older adults.To enhance the quality of your introduction, consider incorporating the following references

https://link.springer.com/article/10.1007/s12126-022-09492-8

https://jrh.gmu.ac.ir/article-1-1745-en.html

https://link.springer.com/article/10.1007/s10943-020-01151-z

https://bmcpsychology.biomedcentral.com/articles/10.1186/s40359-022-00852-2

https://link.springer.com/article/10.1007/s11469-021-00617-9

https://brieflands.com/articles/ijhrba-93481

https://jpcp.uswr.ac.ir/article-1-895-en.html

Method:

Consider including the reliability or validity measures of the chosen items or questions used to measure subjective well-being, social class identity, and Self-rated health. This adds credibility to the chosen instruments.Clarify how missing data were handled, particularly in the exclusion criteria. Specify if imputation techniques were used or if cases with missing values were excluded from the analysis

Discussion

When referring to theoretical frameworks such as Maslow's Hierarchy of Needs theory or social-emotional choice theory, provide a brief explanation of these theories for readers who may not be familiar with themWhen discussing the temporal comparisons between 2018 and 2021, elaborate on the reasons behind the changes observed. Discuss potential societal or economic factors that may have influenced the increase in economic income, SWB, and social class identity.

---

## [Author Response · Author response to Decision Letter 1]

12 Mar 2024

Introduction

• Elaborate more on the broader significance and implications of the study in the context of healthy aging. Highlight the potential contributions of the research to the existing literature and how it addresses gaps in understanding the relationship between SWB, social class identity, and Self-rated health in older adults.

• To enhance the quality of your introduction, consider incorporating the following references

Thanks to the advice of the expert teachers, I have enriched the citation of this paper according to your comments and have cited several references that you have recommended as highly relevant according to the content and purpose of the study.

Method:

• Consider including the reliability or validity measures of the chosen items or questions used to measure subjective well-being, social class identity, and Self-rated health. This adds credibility to the chosen instruments.

Thank you for the expert teacher's opinion. CGSS is a large-scale nationwide social sample survey jointly initiated by Renmin University of China and Hong Kong University of Science and Technology, with the main purpose of understanding the living and employment conditions of urban and rural residents in China as well as their attitudes towards hot issues in the society. The survey aims to explore the living conditions, employment situation and attitudes towards hot issues in society of urban and rural residents. The survey covers 30 provincial-level administrative regions in China, which reflects the real situation of China's society in a more comprehensive and integrated way, and has a certain degree of scientificity and representativeness. Moreover, the CGSS questionnaire has been carefully designed and strictly controlled, and has passed the reliability and validity test, so the questionnaire is scientific and authoritative.

• Clarify how missing data were handled, particularly in the exclusion criteria. Specify if imputation techniques were used or if cases with missing values were excluded from the analysis

Thank you for your comments, I have added and modified them as you requested. Thank you for your comments, I have added and modified them as you requested, see line 176.

Discussion

1. When referring to theoretical frameworks such as Maslow's Hierarchy of Needs theory or social-emotional choice theory, provide a brief explanation of these theories for readers who may not be familiar with them

Thank you for your valuable comments, and I have added a brief description of the relevant theories, as shown in lines 305 and 339.

2. When discussing the temporal comparisons between 2018 and 2021, elaborate on the reasons behind the changes observed. Discuss potential societal or economic factors that may have influenced the increase in economic income, SWB, and social class identity.

Thank you for your suggestion, I had followed your suggestion to analyze the reason behind the change, see line 371.

---

## [Editor Report · Decision Letter 2]

14 Mar 2024

The impact of epidemic infectious diseases on the relationship between subjective well-being and social class identity in older adults: the mediating role of Self-rated health

PONE-D-23-34398R2

Dear Dr. Wei,

We’re pleased to inform you that your manuscript has been judged scientifically suitable for publication and will be formally accepted for publication once it meets all outstanding technical requirements.

Kind regards,

Academic Editor

PLOS ONE

---

## [Editor Report · Acceptance letter]

18 Mar 2024

PONE-D-23-34398R2 

PLOS ONE

Dear Dr. Li, 

I'm pleased to inform you that your manuscript has been deemed suitable for publication in PLOS ONE. Congratulations! Your manuscript is now being handed over to our production team.

Kind regards, 

on behalf of

Dr. Roghieh Nooripour 

Academic Editor

PLOS ONE